# Model-Based Solution for Upgrading Nitrogen Removal for a Full-Scale Municipal Wastewater Treatment Plant with CASS Process

**Mengmeng Liu** [1,2,3,4], **Meixue Chen** [1,3], **Rong Qi** [1], **Dawei Yu** [1,3], **Min Yang** [1,3], **Jiaxi Zheng** [1,3,4], **Yuansong Wei** [1,3,4,*] **and Haizhou Du** [5]

1   State Key Joint Laboratory of Environment Simulation and Pollution Control, Research Center for Eco-Environmental Sciences, Chinese Academy of Sciences, Beijing 100085, China; m_mliu@163.com (M.L.); mxchen@rcees.ac.cn (M.C.); qirong@rcees.ac.cn (R.Q.); dwyu@rcees.ac.cn (D.Y.); 18732186612@163.com (M.Y.); talenthunterjx@163.com (J.Z.)
2   Yangtze River Ecological Environment Engineering Research Center, China Three Gorges Corporation, Beijing 100038, China
3   Laboratory of Water Pollution Control Technology, Research Center for Eco-Environmental Sciences, Chinese Academy of Sciences, Beijing 100085, China
4   University of Chinese Academy of Sciences, Chinese Academy of Sciences, Beijing 100049, China
5   Zhangjiakou Xishan Wastewater Treatment Co., Ltd., Zhangjiakou 076250, China; 18813012008@163.com
*   Correspondence: yswei@rcees.ac.cn

**Abstract:** Aiming at providing cost-effective approach for upgrading the existing municipal wastewater treatment plants in the cold region of China to meet more stringent discharge standards of nitrogen removal, a full-scale sewage treatment plant with the CASS process was selected through focusing on biological process, key equipment and hydrodynamics in bioreactors by the activated sludge model 1 (ASM1) and computational fluid dynamics (CFD) model. Influent COD fractions and the key characteristic parameters ($Y_H$ and $b_H$) of the activated sludge were determined through the respirometry at temperatures of 10 °C and 20 °C, respectively. The layout of submerged agitator installation in the bioreactor of the CASS process was optimized through CFD simulation. The calculation of the average relative deviation (less than 20%) between simulated data and the operation data, demonstrated that the ASM1 model could be reasonably used in the wastewater treatment plant simulation. The upgrade solution based on modelling of ASM1 and CFD was successfully applied in practice, which not only made the effluent COD, $NH_4^+$-N and TN concentrations meet with the discharge standard of Grade I-A, but also reduced the energy consumption by 25% and 16.67% in summer and winter, respectively. After upgrading, microbial diversity increased in both summer and winter, with an especially significant increase of the relative abundance of denitrifying bacteria.

**Keywords:** activated sludge model No.1 (ASM1); sensitivity analysis; full-scale; CASS WWTP; CFD

## 1. Introduction

According to United Nations statistics, by 2025, two-thirds of the world's population may face water shortages [1]. If discharged without treatment, municipal and industrial wastewater will cause damage to the aquatic ecosystem through eutrophication and fish poisoning, as well as adverse effects on human health due to the discharge of pathogenic organisms in sewage to recreational water bodies [2]. However, with proper treatment, water pollution can be minimized and wastewater can be used as a resource for nutrients and recycled water. Therefore, wastewater treatment technology plays a vital role in realizing the sustainable future of human society [3].

Developed countries have experienced the process of "pollution first, treatment later" of the water environment. Domestic sewage treatment developed rapidly after the 1970s. By the end of the 1990s, the average domestic sewage treatment rate in developed countries

had reached a relatively high level of over 80% [4]. Among them, New Zealand, Singapore, and Northern Europe have basically achieved 100% collection and processing rates. In developed countries, sewage treatment facilities are mainly small and medium-sized. For example, there are more than 14,780 sewage treatment plants nationwide in the United States, with an average daily processing water volume of about 6000 tons per plant, of which small and medium-sized sewage treatment plants with a capacity of less than 50,000 tons per day account for about 85% of the total [5]. The United States is currently the country with the largest number of sewage treatment plants in the world, with an average of 1 in 5000 people, 78% of which are secondary biological treatment plants; the United Kingdom has about 8000 treatment plants with an average of 1 in 7000 people, almost all of which are secondary biological treatment plants. There are about 630 urban wastewater treatment plants in Japan, with an average of 1 in 200,000 people, but secondary treatment plants and advanced treatment plants account for 98.6%; Sweden is currently the country with the most popular sewage treatment facilities, with a sewer penetration rate of over 99%, with an average of 1 in 5000 people, of which 91% are secondary biological treatment plants [6].

With increasing emphasis on China's water environment protection, more stringent regulations and standards have been legislated governing the discharge of organic pollutant, nitrogen and phosphorus in effluent of municipal wastewater treatment plants (WWTPs). Therefore, there is a significant need to upgrade the existing municipal WWTPs to meet the strict discharge standards. For example, the Action Plan for Water Pollution Prevention and Control issued in 2015 by China [7] requests effluent from all WWTPs into the receiving water bodies to meet with requirements of the Grade I-A of the Discharge Standard of Pollutants for Municipal Wastewater Treatment Plant (GB 18918–2002), e.g., chemical oxygen demand (COD), total nitrogen (TN), total phosphorus (TP) concentration limits of effluent should be upgraded to 50 mg/L, 15 mg/L and 0.5 mg/L of the Grade I-A from 60 mg/L, 20 mg/L and 1.0 mg/L of the Grade I-B, respectively. It was reported that the total municipal wastewater discharged in China was 49.24 billion tons in 2017, increased by 30.02% and 5.53% compared with 2010 and 2015, respectively, along with rapid urbanization [8]. Among the over 4000 municipal WWTPs in China, the small and mid-scale municipal WWTPs with the design capacity less than 50,000 m$^3$/d accounted for 83.3% in 2017 [9], most of which need to be upgraded or optimized to meet the discharge requirements of Grade I-A. Generally, the upgrading of the existing sewage treatment plants can be considered from two perspectives: one is only the effluent water quality of WWTPs, the other is both the effluent water quality and the design capacity of WWTPs [10], and the most important goal of upgrading existing municipal wastewater treatment plants is to ensure high treatment efficiency to meet wastewater quality standards, while keeping investment and operating costs to a minimum by focusing on energy saving and cost reduction methods of bioprocesses, key equipment and hydrodynamics in bioreactors. However one of major challenges of upgrading the existing municipal WWTPs in China, especially those in the small and mid-size cities & towns of the cold region, to meet the Grade I-A standards of GB 18918–2002 is the TN limit of 10 mg/L in the effluent while keeping the investment and operating cost as low as possible [11,12].

Wastewater treatment can be divided into physical, chemical and biological treatment methods based on the type of water quality [13]. According to the degree of treatment, they can also be divided into primary, secondary and tertiary treatment methods [14]. The physical treatment method of urban sewage is a method that uses physical action to separate and remove pollutants in sewage [15], including screening and interception, gravity separation, centrifugal separation. Chemical treatment methods are rarely used in urban sewage treatment, and generally involve other chemical methods in urban water supply treatment, such as neutralization, oxidation-reduction, ion exchange, and electrolysis, which are mainly used for industrial wastewater treatment [16]. Chemical method must be used in conjunction with the physical method. Before chemical treatment, precipitation and filtration are often used as pre-treatment; in some cases, physical means such as

precipitation and filtration are required as post-treatment of chemical treatment. Biological treatment is a method that uses the metabolism of microorganisms to remove organic substances in sewage. Commonly used are activated sludge method, biofilm method, as well as oxidation pond and sewage land treatment method [17].

It is well known that the activated sludge processes are the most applied biological approach for nutrient removal for municipal wastewater treatment in the world. Traditionally, municipal WWTPs have been designed and operated using empirical steady-state equations or 'rules of thumb', introducing conservative safety factors that have led to the over-dimensioned, expensive construction and unstable functioning plants [18]. Different modifications have been thus proposed for the conventional activated sludge processes to meet the strict effluent standards for COD, biological oxygen demand (BOD), nitrogen and phosphorus [19].

Due to the nonlinear dynamics and uncertainty, the internal process dynamics have multiple time scales and multivariable structures, so activated sludge wastewater treatment plants can be classified as complex systems, and such traditional approaches cannot meet increasing requirements for municipal WWTPs. Process evaluation, design optimization, and cost analysis can be performed by simulation and modeling tools [20–22]. Numerical simulation based on activated sludge models is getting an increasing attention, and these models are incorporated in commercial simulation packages such as BioWin (EnviroSim Associates, Flamborough, Ontario, Canada), GPS-X (Hydromantis Inc., Ontario, Canada) and WEST (HEMMIS Inc., East Flanders, Belgium and DHI Inc, Copenhagen, Denmark) for engineering practice [23,24]. Numerical simulation has been widely regarded as helpful tools for evaluation, diagnosing and optimizing WWTPs' design, and operation and control, comparison and selection of the biological treatment processes as well as upgrade of the existing WWTPs [25–27]. A series of activated sludge models (i.e., ASM1, ASM2, ASM2d, ASM3) formulated and introduced by IWA (International Water Association) have been considered as indispensable solutions in correlating the complexity of the activated sludge process and the prediction of biological treatment efficiency under dynamic conditions [28–30].

The ASM1 is the primary version, which is a structured model based on Monod kinetics to realistically predict the performance of carbon oxidation, nitrification and denitrification in activated sludge systems under aerobic and anoxic conditions. Among these available models, ASM1 has been considered as a reliable reference model due to its most widely applications for the academic and operational fields [31–33].

Nevertheless, one of the major limitations for a more widespread application of ASM is to choose a set of related parameters that are essential for achieving a good prediction of the model used [34]. In the neutral and relatively constant conditions, there are many environmental factors that influence the dynamics and stoichiometric parameters of ASM1 models, but influent characteristics and temperature are the two most common environmental factors [30,35]. Most of the parameters are affected by the specific components of the influent [36–38], which either promote or inhibit the values of kinetic parameters and stoichiometric coefficients.

However, the characteristics of the influent fractions can change in different seasons during one year [39]. Different seasons and influent characteristics lead to different kinetic parameters. A small water temperature range (low temperature, medium temperature, high temperature) generally causes the rate coefficient changing (e.g., $\mu_{H,max}$, $b_A$ or $k_h$). Moreover, almost all kinetic parameters are affected by water temperature, so the influence of temperature should be considered when setting values (Henze et al., 2000). Although many literatures have proposed different water temperature correction factors [40–43], most of them are not universally suitable for all the researches. In a word, it is necessary to simultaneously determine the concentration of influent characteristics and the key kinetic parameters at different temperatures so the models can be accurately applied to the design, operation and upgrade of existing wastewater treatment systems [30].

Flow and hydrodynamic characteristics play an important role in the stable operation of an activated sludge processes system, especially for the denitrification process within a limited residence time. While in most of the anaerobic or anoxic tanks, the submersible agitators were installed on the basis of general empirical guidelines, operators' experience or 'rules of thumb', which cannot guarantee the evenly mixing of the flow field. Improper installation of the agitators can also cause the damage to the blades. CFD is a powerful tool to simulate the hydrodynamics and mass transfer, and has become increasingly popular in optimizing design and operation of WWTPs [44–48], therefore such a study of the agitators' location and position in the bioreactor has become easier and less expensive.

In China, the Sequencing Batch Reactor (SBR) and its variant processes was the third widely used activated sludge process at 17.19% of municipal WWTPs, while the first two were the oxidation ditch (OD) process at 29.21% and the anaerobic–anoxic–oxic ($A^2$/O) process at 25.45% [49]. The CASS process is a variant of the SBR process, in which nitrogen is removed mainly by simultaneous nitrification and denitrification [50]. The CASS process has been widely used in municipal and industrial WWTPs, e.g., over 400 small and mid-scale municipal WWTPs, especially those in cold regions of northern China because of its configuration flexibility, operational simplicity, low construction and maintenance costs, and simultaneous removal of nitrogen and phosphorus [51]. It is obvious that treatment performance of small and mid-scale municipal WWTPs in the cold regions of northern China, especially the TN removal, has significant difference in summer and winter, e.g., unstable TN removal and TN concentration of effluent usually does not meet the discharge standard in winter [52]. Considering key factors such as large fluctuations in water quality and quantity of influent, unstable performance of wastewater treatment in winter, selection of new process or optimization of the existing process, constraints of existing structure and field, and limits of capital & operational costs, it has been suggested that the mathematical ASM model will be a very powerful and cost-effective tool for upgrading, designing, operating and optimizing existing small and medium-scale sewage treatment plants in the cold regions of northern China.

Thus, a hypothesis in this study is that the model-based upgrading of an existing full-scale WWTP is feasible through optimizing the existing biological process to comply with TN of effluent quality criteria while keeping the capital and operational costs as low as possible. A full-scale municipal WWTP with the CASS process at a design capacity of 20,000 $m^3$/day, located in Zhangjiakou, Hebei Province, was selected to upgrade TN removal on the basis of ASM1 modelling and optimization, as well as CFD simulation and optimization for the agitators' layout. In this study, the modelling, simulation and optimization of this full-scale CASS process WWTP was carried out by ASM1 using a commercial software package GPS-X 8.0 (Hydromantis Inc., Ontario, Canada) to evaluate the treatment performance and diagnose the bottlenecks of operation in summer and winter, determine the influent COD fractions and calibrate the model kinetic parameters. Then, an upgrade solution of this WWTP with different operational strategies and multiple optimization alternatives, as well as the optimized location and position of the agitators in CASS tank with the aid of CFD simulation, were made and carried out to make effluent quality meet the requirements of Grade I-A of the GB 18918–2002. In addition, treatment performance (especially TN removal) and energy consumption of this WWTP were compared before and after the upgrading of this WWTP, and the evolution of the microbial community of the CASS process with seasonal changes was also investigated based on high-throughput 16S rRNA gene sequencing analysis.

## 2. Materials and Methods

### 2.1. Full-Scale Wastewater Treatment Plant and CASS Process

In this study, a full-scale municipal WWTP with the CASS process located in north-west of Hebei province (North China) was selected, which design capacity is 20,000 $m^3$ of domestic wastewater per day [53]. This WWTP consists of four CASS systems arranged in parallel, mainly including a mechanical treatment unit of primary settling tank to

remove floating and settleable solids, a biological treatment unit with activated sludge process for COD and nutrients removals, a sludge treatment unit of dewatering. Before upgrading, the bioreactor of the CASS consisted of a pre-denitrification anoxic zone (L × W × H = 4.3 m × 17.6 m × 5.6 m, with effective volume of 423.8 m$^3$), in which there are two submersible agitators installed on the opposite corners to make the anoxic tank in good mixing performance. Without submersible agitators, the aeration in the nitrification aerobic zone (L × W × H = 34 m × 17.6 m × 6.5 m, effective depth 5.5 m with working volume of 3,291.2 m$^3$) was obtained with fine-pore air diffusers located at the bottom of the bioreactor to make the DO concentrations at 2–3 mg·L$^{-1}$. A schematic flow diagram of the biological step of the CASS system was shown in Figure 1a.

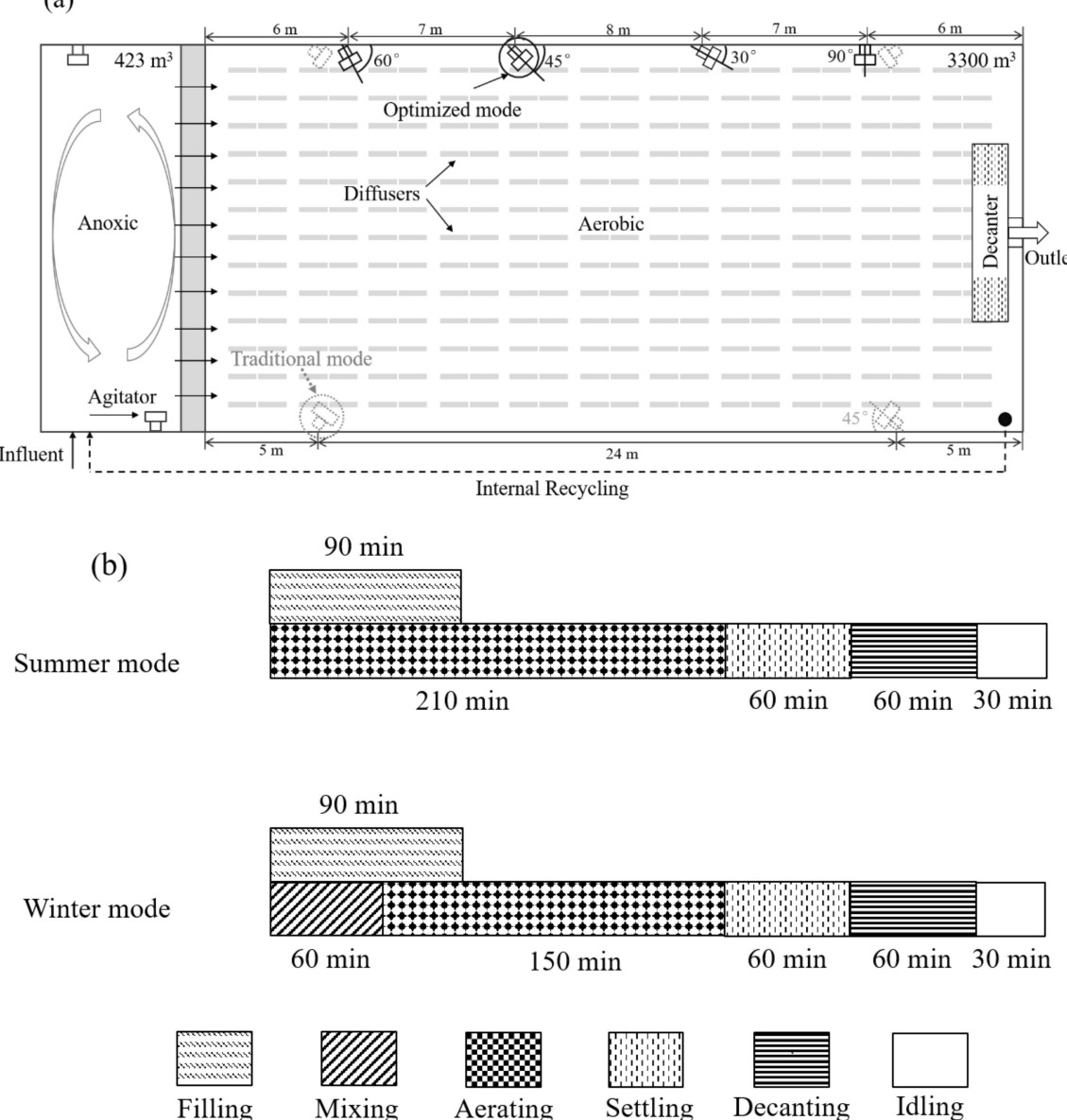

**Figure 1.** Schematic description of CASS and the two operation modes in summer and winter. (**a**) Top view of biological tank; (**b**) Operation strategies.

The CASS process was operated under two different strategies in summer (June, July and August) and winter (December, January and February), respectively (Figure 1b). Before upgrading, each CASS cycle was designed for 6 h, while the effluent cannot meet discharge standard of TN limit at 15 mg·L$^{-1}$ (GB18918–2002) in winter (Figure S1). In summer mode,

90 min was allocated to batch feeding without the submersible agitator operation and most of the denitrification process takes place in the front anoxic zone. The nitrification of ammonia nitrogen was achieved by aeration for 210 min. For the last phase, settling and decant time were the same for 60 min, and 30 min for idling phase. The only difference between the summer mode and winter mode was the first two phases. In winter mode, the first two phases consist of 60 min for mixing during the time of 90 min of filling, and the aeration phase was shortened to 150 min. These two kinds of operation strategies depended on the influent characteristics as shown in Table 1. From the beginning of September 2018, the TN of effluent must meet Grade I-A discharge standard of GB18918–2002, while the residence time of the pre-denitrification anoxic zone is too short to meet the standard nitrogen emission. Because the wastewater filling time of a single CASS system is 90 min, in order to ensure the continuous water supply of the four series as a whole, sometimes the idle period is increased in the operation cycle.

**Table 1.** Operating parameters and influent characteristics of the CASS process (data from June, 2018 and December, 2018).

| Parameter. | Operation Mode | Mean $\pm$ SD | Minimum | Maximum |
|---|---|---|---|---|
| Volume of wastewater treatment ($m^3 \cdot d^{-1}$) | Summer | 18,600 $\pm$ 1530 | 16,500 | 19,800 |
| | Winter | 14,300 $\pm$ 2840 | 12,130 | 16,900 |
| Wastewater temperature (°C) | Summer | 20.3 $\pm$ 0.74 | 18.5 | 21.8 |
| | Winter | 9.8 $\pm$ 0.68 | 8.1 | 11.3 |
| SRT (d) | Summer | 21 $\pm$ 0.6 | 20.2 | 22.3 |
| | Winter | 24 $\pm$ 0.9 | 22.2 | 25.7 |
| RAS flow rate ($m^3 \cdot d^{-1}$) | Summer | 9525 $\pm$ 1010 | 8991 | 10,789 |
| | Winter | 12,525 $\pm$ 1324 | 10,991 | 15,317 |
| pH | Summer | 7.7 $\pm$ 0.19 | 7.26 | 7.91 |
| | Winter | 7.5 $\pm$ 0.12 | 7.26 | 7.93 |
| SS ($mg \cdot L^{-1}$) | Summer | 126.48 $\pm$ 24.24 | 67 | 163 |
| | Winter | 155.39 $\pm$ 0.50.24 | 67 | 429 |
| COD ($mg\ O_2 \cdot L^{-1}$) | Summer | 341.45 $\pm$ 61.2 | 193 | 549 |
| | Winter | 446.59 $\pm$ 79.74 | 318 | 689.16 |
| BOD$_5$ ($mg\ O_2 \cdot L^{-1}$) | Summer | 157.97 $\pm$ 37.24 | 82 | 254 |
| | Winter | 217.3 $\pm$ 38.37 | 129.92 | 396 |
| TN ($mg\ N \cdot L^{-1}$) | Summer | 56.97 $\pm$ 12.54 | 43.8 | 79.3 |
| | Winter | 90.70 $\pm$ 8.62 | 67.81 | 125.89 |
| NH$_4^+$-N ($mg\ N \cdot L^{-1}$) | Summer | 48.81 $\pm$ 11.3 | 40.91 | 59.18 |
| | Winter | 71.39 $\pm$ 11.01 | 47.60 | 88.70 |
| TP ($mg\ P \cdot L^{-1}$) | Summer | 6.9 $\pm$ 2.63 | 2.46 | 14.3 |
| | Winter | 7.02 $\pm$ 2.29 | 5.88 | 8.42 |
| C/N ratio | Summer | 5.74 $\pm$ 1.56 | 4.79 | 9.17 |
| | Winter | 5.58 $\pm$ 1.11 | 5.09 | 8.32 |

To maintain the biomass, the returned sludge, nearly 50% of the influent flow rate in summer mode, while 100% in winter, from the end of SBR zone was recirculated into the anoxic basin. The MLSS was controlled at about 4000 mg·L$^{-1}$ in the main SBR zone after filling phase, while approximately 5200 mg·L$^{-1}$ after decanting with a volumetric exchange ratio of 22.73% for each cycle. The SRT was controlled at nearly 21 d by discharging an appropriate amount of sludge.

## 2.2. ASM1 Modelling

The ASM1 was used in this study for its good description of the activated sludge process [41,54]. Firstly, the influent COD fraction, heterotrophic yield, $Y_H$, and heterotrophic decay rate, $b_H$, were estimated by the respirometry test in summer (20 °C) and winter

(10 °C), respectively. Secondly, the essential kinetic and stoichiometric model parameters were determined through sensitive analysis. Thirdly, the results of the simulation based on the field investigation data of this WWTP were compared with the actual treated effluent water quality. Fourthly, an upgrade solution with different operational strategies and multiple optimization alternatives, as well as the optimized location of the stirrers in CASS tank with the aid of CFD simulation, were put forward and carried out to achieve better effluent quality.

### 2.3. Respirometry Test

Respirometry is the useful method in diagnose the COD fractions and some key model parameters [37,41]. This method has been widely used in the ASM modelling [55–57]. The respirometry experiments were carried out in a 2 L closed vessel which was magnetic stirred to make the proportional liquor well-mixed, and the dissolved oxygen (DO) electrode (Oxi 3205, WTW, Berlin, Germany) was fixed in the vessel to monitor the DO data intermittently. The temperature of these experiments was kept at 20 °C for summer samples while 10 °C for winter samples throughout the respirometry test period.

The activated sludge samples taken from the CASS system in different seasons (summer and winter in 2018) were aerated for 24 h before use to ensure the endogenous state at the beginning of the experiments. After aeration, in order to remove the external COD, the static sedimentary sludge was then washed for three times with distilled water. Six hundred mL wastewater and 400 mL washed sludge were mixed in a closed vessel according to the actual system operation F/M value. All tests were carried out under the conditions of adding 20 mg·L$^{-1}$ of nitrification inhibitor (Allylthiourea, ATU, Shanghai, China) to limit the oxygen consumption by nitration. DO concentration in the vessel was recorded intermittently with the dissolved oxygen electrode (Oxi 340i, WTW, Berlin, Germany). The first step was to rapidly increase the DO of the mixture up to 6 mg·L$^{-1}$ with strong aeration followed by a decrease phase of DO concentration to 2 mg·L$^{-1}$ by turning off the air pump. The data sampling frequency is once every 10 s, which is relatively high in the initial stage for the high OUR rate due to $S_s$ degradation. The dissolved oxygen in the reactor will drop to 2 mg·L$^{-1}$ in a short period of time (<5 min). At this time, the data reading will be suspended, the aerator will be used to oxygenate the mixture system in the vessel. The aerator was turned off after the DO was above 6 mg/L again. Then, the change of the DO in the reactor was recorded intermittently. The above steps were continued until the value of the oxygen consumption rate becomes constant. Thus, the OUR of the sludge can be calculated by the slope of the DO concentration reduction during period in which the aeration was turned off.

### 2.3.1. Heterotrophic Yield $Y_H$

The actual heterotrophic yield coefficient, $Y_H$, which must be known before determining the soluble rapidly biodegradable organic, $S_S$ [37,58]. It was evaluated by a respirometry test in which four different concentrations of fully biodegradable organic substrate were added to the our system in which the sludge was in endogenous phase [59]. Sodium acetate (AP-10018818, Sinopharm Chemical Reagent Co., Ltd., Shanghai, China) was used in this test since it was the external carbon source added during the anoxic periods of the case studied [59,60].This test was operated at 20 °C and 10 °C, respectively, and a pH value of 7.5 ± 0.1 with a low concentration of MLVSS, which provided a suitable low OUR that improved the $Y_H$ assessment.

According to the literature [60], the heterotrophic yields can be calculated from the following equation:

$$OC = (1 - Y_H) \cdot S_{Ace} \tag{1}$$

where OC is the accumulated exogenous oxygen consumption, $S_{Ace}$ is the concentration of the added sodium acetate. The OC in summer and winter was calculated according to the added biodegradable organic substrate concentration, respectively.

### 2.3.2. Heterotrophic Decay Rate $b_H$

The heterotrophic decay rate, $b_H$, is critical for the prediction of sludge production and oxygen demand [30]. Therefore, it must be determined based on the activated sludge used in the OUR test of $b_H$.

Since heterotrophic bacteria were in the endogenous respiratory stage after 24 h of aeration, OUR can only be caused by the microbial auto-oxidation. There was a linear relationship between ln (OUR) and time $t$, the slope of the curve is a negative traditional decay rate which follows the next formula [41]:

$$\ln(\text{OUR}) = -K_d t + \ln(f_{vC} K_d X_{0B,H}) \tag{2}$$

where $f_{vC}$ is a proportional constant; $K_d$ is global attenuation coefficient, $X_{0B,H}$ is initial concentration of active heterotrophic biomass (mg/L)

The $b_H$ can be calculated from the next equation which is related to $K_d$ [30]:

$$b_H = \frac{K_d}{1 - Y_H(1 - f_p)} \tag{3}$$

For each summer and winter, 1500 mL activated sludge taken from the CASS system of this WWTP was put in a batch reactor, then washed with the distilled water after 24 h aeration, and the OUR was further measured for multiple times in 4 days [41]. During the experiments, pH and temperature were controlled the same as the values in summer and winter, respectively.

### 2.3.3. COD Fractionation of Influent

In ASM1 model, the constituent elements of total COD were presented by equation:

$$\text{COD} = S_I + S_S + X_I + X_S \tag{4}$$

where:

$S_I$–soluble inert substrates, g $O_2$ m$^{-3}$;

$S_S$–soluble readily biodegradable substrates, g $O_2$ m$^{-3}$;

$X_I$–inert particulate organic material, g $O_2$ m$^{-3}$;

$X_S$–particulate slowly biodegradable substrates, g $O_2$ m$^{-3}$;

According to the Equation (4), the influent COD fractions can be divided into the above four components and be used as the influent simulation data for the ASM1 model.

The soluble COD value (SCOD) of sewage is usually defined as the COD through a 0.45 μm microfiltration membrane. According to the literature [61], the floc in the zinc sulfate coagulation filtration method has a small adsorption amount to the SCOD, and the supernatant COD is close to the truly solution COD. Therefore, the steps of the physical and chemical separation method to determine the SCOD were as follows: Firstly, 10 mL of 100 g·L$^{-1}$ zinc sulfate was added to 1 L of the wastewater sample to be tested, and the pH was adjusted to about 10.5 with 6 mol·L$^{-1}$ NaOH under the rapid stirring of the magnetic stirrer; Secondly, the liquor was mixed at high speed (120 r·min$^{-1}$) for 1 min and low speed (60 r·min$^{-1}$) for 5 min by magnetic stirrer, and then static settlement for 15 min; At last, conventional disposable filters with pore size of 0.45 μm (Durapore$^{®}$ Membrane Filter, polyvinylidene fluoride (PVDF), New York, USA) were used for sequential filtration after sedimentation. The COD value of the filtrate is the wastewater SCOD [36].

The OUR curve of the mixture could be divided into three segments. In S1, t < $t_1$, the OUR value sharply decreased because the substrate was easily biodegradable; in S2, $t_1$ < t < $t_2$, OUR slowly decreases, the rate of which is controlled by the substrate provided by the slow degradation of the substrate $X_S$ in the wastewater; in S3, t > $t_2$, OUR is almost constant and is maintained at a relatively low level, which can be considered as the oxygen consumption of endogenous respiration of the original heterotrophic microorgan-

isms before mixing with wastewater. As a result, the $S_S$ and $X_S$ could be calculated from the partial OUR curves in S1 and S2 individually.

To get a clear description of the OUR curves for COD fractions, before the respiroetric tests, the volume of the wastewater, $V_W$, and the activated sludge, $V_S$, should be determined to get a suitable F/M ratio (the ratio between the SCOD of influent value and the MLVSS). The suggested F/M ratio for the OUR tests was between 0.01 and 0.2 mg COD/mg VSS [38]. For each season's OUR test, triplicate experiments were performed to get a good repeatability. $S_S$ and $X_S$ could be calculated by the following equations:

$$S_S = \frac{V_W + V_S}{V_W} \cdot \frac{1}{1 - Y_H} \int_0^{t_1} (OUR_{tot} - OUR_{X_S}) \cdot dt \tag{5}$$

$$X_S = \frac{V_W + V_S}{V_W} \cdot \frac{1}{1 - Y_H} \int_{t_1}^{t_2} (OUR_{X_S} - OUR_{ER}) \cdot dt \tag{6}$$

where $OUR_{tot}$ is total oxygen uptake rate (mg/(L·d$^{-1}$)), $OUR_{X_s}$ is oxygen uptake rate of $X_S$ consumption and endogenous respiration (mg/(L·d$^{-1}$)), $OUR_{ER}$ is oxygen uptake rate of endogenous respiration (mg/ (L·d$^{-1}$))

### 2.4. Model Calibration and Validation Strategy

In this study, the GPS-X software (Hydromantis Inc., Ontario, Canada) with ASM1 model was used to simulate the CASS system. Two key parameters ($Y_H$, $b_H$) were measured based on the respirometry batch experiment, other model parameters were mainly corrected according to the sensitivity analysis. The input data for the simulations came from the CASS WWTP and comprised the period of June 2018 for the calibration and July 2018 for the validation for summer simulation, December 2018 for calibration and January 2019 for validation for winter simulation, respectively. Table 1 lists the operating parameters and wastewater characteristic values used for model calibration and verification.

The validated model was used for simulation and optimization to achieve complete denitrification in wastewater, thereby providing upgrade solutions to meet TN emission standards and reduce operating costs. Optimization was completed by running multiple simulations on different operating scenarios, such as reducing aeration time and testing different operation modes in the CASS tanks, increasing anoxic time to perform nitrification-denitrification. The optimized operation strategies were applied to the CASS operation in practice from August 2018 to February 2019, respectively, to acquire the optimized operation data from the WWTP.

### 2.5. Sensitivity Analysis

Sensitivity analysis can assess to what extent the parameters used in model calibration affect the output of the model [62]. Sensitivity analysis before model calibration is necessary to evaluate important parameters [63]. According to EPA guidelines [64], the sensitivity coefficient ($S_{i,j}$) is defined as a ratio of the percentage change in the output variable ($y_i$) to a 10% change in the input variable ($x_i$):

$$S_{i,j} = \frac{\Delta y_i \ / \ y_i}{\Delta x_j \ / \ x_j} \tag{7}$$

In this study, all model coefficients (including kinetic coefficients and stoichiometric coefficients) were changed by 10% in the simulation. The determination of key parameters during calibration depends on the sensitivity of the model output to these parameters. The effect of parameters on the model output can be explained as: (1) < 0.25 means that the parameter has no significant effect on the model output, (2) $0.25 \leq |S_{i,j}| < 1$ means that the parameter has an effect; (3) $1 \leq |S_{i,j}| < 2$ indicates that the parameters are very influential; (4) $|S_{i,j}| \geq 2$ indicates that the parameters are extremely influential [65].

### 2.6. CFD Modelling

In full-scale WWTPs, submerged agitators are always used to control high-flow mixing, and it is generally known that the single performance parameter thrust (F) is the basis of the design of the agitation system and the positioning principle of a series of agitators [57]. CFD can be used to model the mixer in detail, but it is too complicated to be included in a wide range of plant models, instead, at the same agitators geometrical location, a simile was used to contain the mechanical momentum added by the agitators in the system as the momentum source M (kg·m$^{-2}$·s$^{-2}$) [66]. The detail description used in this section was referred to the literature [67].

In order to evaluate the effect of the two different installation layouts of the agitators on the fluid dynamics and its influence on the kinetic model, a CFD three-dimensional single-phase method was performed by ANSYS® Academic Research Release 18.0 software (ANSYS, Inc., New York, NY, USA) which was used as the CFD modelling in this study. The different simulations described in this section were run on ANSYS-Fluent against traditional and modified configurations to achieve these goals. The second-order upwind and PRESTO scheme, which was suitable for swirling flow, were used for discrete spatial derivatives and discrete pressure, respectively. The semi-implicit method of pressure link equation (SIMPLE) is used to realize the coupling of speed and pressure. At the beginning of the simulation, the under-relaxation factor is reduced to maintain stability and avoid solution divergence. When the proportional residual continuity drops below $1 \times 10^{-4}$ and the velocity and turbulence drop below $1 \times 10^{-5}$, the solution is considered to be fully converged.

### 2.7. High-Throughput 16S rRNA Gene Sequencing

In order to reveal the impact of microbial population changes on the performance of the CASS system, activated sludge samples collected in four seasons were selected as samples for 16S rRNA gene amplicon sequencing. These samples were processed in order to perform DNA extraction and 16S rRNA gene PCR, which was followed by the amplification and purification of PCR products. To extract DNA, 2 mL sludge sample was mixed with the DNA extraction kit according to the manufacturer's instructions. In order to ensure the accuracy of the extraction, four times of DNA samples were extracted from each sample and then mixed evenly. The DNA extracts were purified and stored at –20 °C until the analysis. For PCR amplification, bacterial 16S rRNA fragments were amplified by adding different eight-base barcodes to the forward primer (5′-GTGCCAGCMGCCGCGGTAA-3′) and reverse primer (5′- GCCAGCMGCCGCGGTAA-3′) of each sample.

After PCR amplification of 16S rRNA gene, the amplicons were purified using SanPrep DNA gel extraction kit. Before transferring the purified PCR product to the sequencing analysis step, it is quantified. Finally, Shanghai Sangong Biotechnology Co., Ltd. (Shanghai, China) performed high-throughput sequencing on the Illumina sequencing platform

## 3. Results and Discussion

### 3.1. Assessment of the CASS Process Performance

The $NH_4^+$-N and TN removals by the CASS system showed obvious seasonal changes, with higher nitrification and denitrification capacity in summer while lower in winter (Figure S1). The effluent TN and $NH_4^+$-N concentrations were $16.8 \pm 2.5$ mg·L$^{-1}$ and $1.25 \pm 0.46$ in summer, respectively, while they were $18.6 \pm 3.5$ mg·L$^{-1}$ and $3.3 \pm 1.5$ mg·L$^{-1}$ respectively, during winter. It should be noted that these TN values exceeded the discharge limit of 15 mg·L$^{-1}$ during both summer and winter.

In order to better understand nitrogen transformation characteristics in CASS system in summer and winter, the variations of $NH_4^+$-N, $NO_3^-$-N, TN concentrations and COD in the main CASS reaction zone were analyzed at a time interval of 30 min and their results are shown in Figure S2. The $NO_3^-$-N concentration decreased greatly at the beginning of the operation filling phase, even though aeration is turned on at the same time (Figure S2a). This may be mainly due to the dilution of the residual water in the watershed by the

incoming water. From Figure S2a,b, no matter in summer or winter, during the aeration phase, the nitrification occurred along with the start of aeration. However, there were less oxygen for nitrification process when it was used for COD degradation. The rapid decline of $NH_4^+$-N occurred obviously when the COD concentration was below 40 mg·$L^{-1}$. Furthermore, the $NH_4^+$-N concentration meets the Grade I-A discharge standard of GB 18918–2002 before the end of aeration. As the aeration stopped, the nitrate concentration stabilized, indicating that no denitrification occurred. This may be due to the absence of exogenous carbon during the decanting and settling phases.

The above onsite experimental finding confirmed that the residence time of the pre-denitrification anoxic zone was too short to meet the discharge standard of nitrogen, and it is necessary to extend the mixing time before aeration to make full use of carbon sources for denitrification and reduce aeration time.

### 3.2. Determination of Model Parameters and Influent COD Fractions

3.2.1. Heterotrophic Yield ($Y_H$) and Heterotrophic Decay Rate ($b_H$)

Heterotrophic Yield ($Y_H$)

The actual heterotrophic yield coefficient, $Y_H$ must be known before determining the soluble rapidly biodegradable organic, $S_S$. This parameter not only affects the estimation of sludge production and oxygen demand, but also affects the value of other parameters whose determination requires a value for $Y_H$, like $b_H$.

According to the formula (1), OC must be zero when there is no substrate added. The regression lines for the OC as a function of the start concentration of substrate are therefore forced through (0, 0). This procedure means that one yield coefficient is assumed to be valid for the whole area of examined substrate concentrations. The plot of the OC versus the $S_{Ace}$ enabled the calculation of (1-$Y_H$) as the slope (Figure 2a).

Experimental evaluation of model parameters $Y_H$ at different temperatures was shown in Figure 2a. The calculated $Y_H$ in different seasons were 0.663 and 0.615, respectively, which revealed the calculation of $Y_H$ was a little affected by the temperature, the lower the temperature, the slower the rate of heterotrophic bacteria.

Heterotrophic Decay Rate ($b_H$)

The decay coefficient $b_H$ of heterotrophic biomass was determined with linear death according to established procedures [68]. In this study, in the four-day test described in Section 2.3.2, the OUR value was calculated every 12 h, and the graph of ln (OUR) versus time is shown in Figure 2b. The calculated $K_d$, the slope of the regressed equation of the curve, were −0.1988/day and −0.1526/day, respectively. After substituting $K_d$ into Equation (3), the calculated $b_H$ of the activated sludge in summer and winter were 0.51/day and 0.28/day, respectively.

3.2.2. COD Fractions of Influent

In this study, the method described in Section 2.3 was used to analyze the influent wastewater sampled from the CASS system in summer and winter, and the methods for obtaining four COD fractions in wastewater were introduced in detail.

Determination of $S_I$ Fraction

$S_I$ is defined as soluble inert organic matter in wastewater. After introducing the mixed liquid containing activated sludge and wastewater treatment plant wastewater into the experimental equipment in Section 2.3, the COD value of the wastewater is analyzed every 8 h during the 48-h aeration period until the COD is constant. Then, the soluble inert organic matter $S_I$ was calculated which were 36.52 mg·$L^{-1}$ (TCOD = 342.33 mg·$L^{-1}$) in summer and 29.93 mg·$L^{-1}$ (TCOD = 447.33 mg·$L^{-1}$) in winter, respectively (Table 2). From the results of $S_I$, the soluble non-biodegradable substance is a little lower in winter than that in summer.

(a)

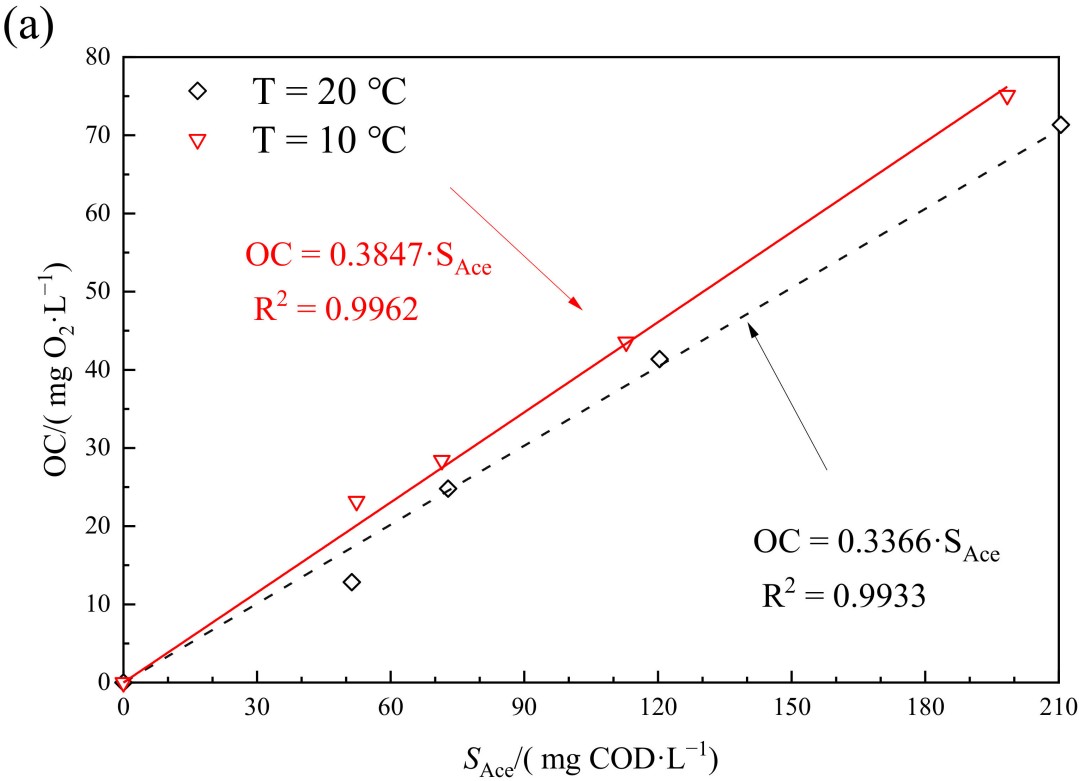

(b)

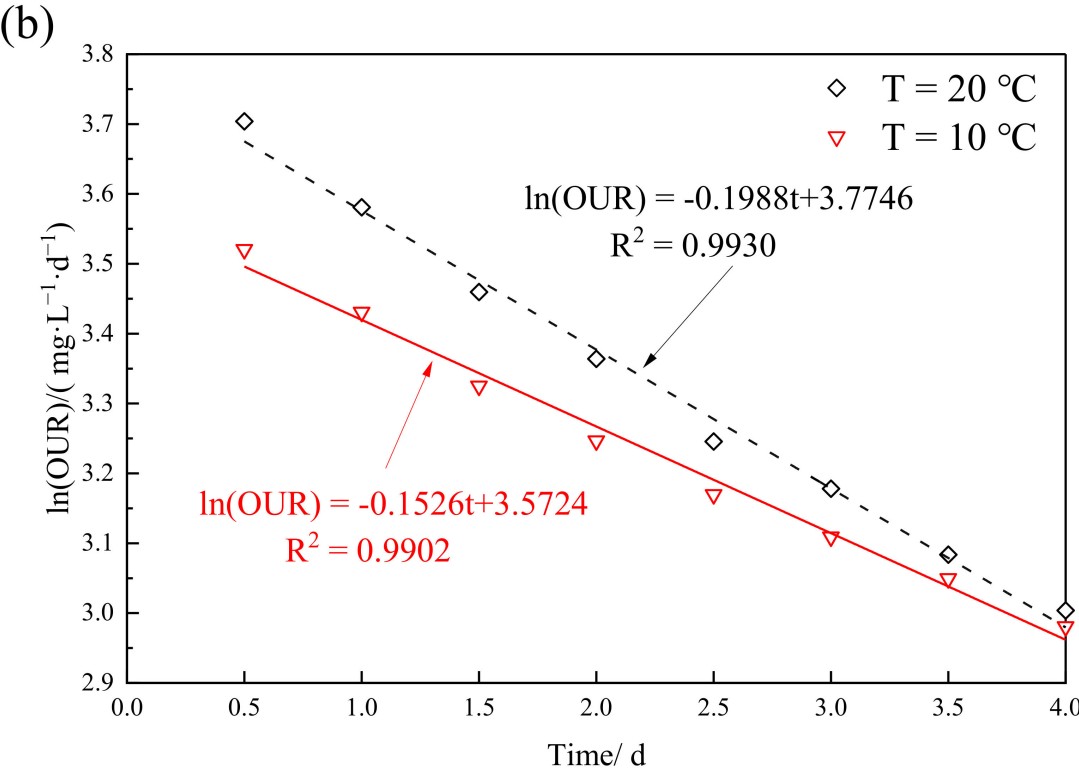

**Figure 2.** Experimental evaluation of model parameters (**a**) $Y_H$ and (**b**) $b_H$ at different temperatures.

**Table 2.** COD fractions of the CASS process in summer and winter.

| Sources | Sample ID | | TCOD | COD Fractions | | | | | | | |
|---|---|---|---|---|---|---|---|---|---|---|---|
| | | | | $S_S$ | | $S_I$ | | $X_S$ | | $X_I$ | |
| | | | mg·L$^{-1}$ | mg·L$^{-1}$ | % | mg·L$^{-1}$ | % | mg·L$^{-1}$ | % | mg·L$^{-1}$ | % |
| This study | Summer | 1 | 356.00 | 117.91 | 33.12 | 40.23 | 11.30 | 165.18 | 46.40 | 32.68 | 9.18 |
| | | 2 | 328.00 | 98.99 | 30.18 | 33.75 | 10.29 | 161.70 | 49.30 | 33.55 | 10.23 |
| | | 3 | 343.00 | 110.99 | 32.36 | 35.57 | 10.37 | 168.69 | 49.18 | 27.75 | 8.09 |
| | | Ave. | 342.33 | 109.30 | 31.89 | 36.52 | 10.65 | 165.19 | 48.29 | 31.33 | 9.17 |
| | Winter | 1 | 440.00 | 93.94 | 21.35 | 26.84 | 6.10 | 261.10 | 59.34 | 58.12 | 13.21 |
| | | 2 | 482.00 | 98.76 | 20.49 | 33.35 | 6.92 | 277.82 | 57.64 | 72.06 | 14.95 |
| | | 3 | 420.00 | 98.95 | 23.56 | 29.61 | 7.05 | 231.76 | 55.18 | 59.68 | 14.21 |
| | | Ave. | 447.33 | 97.22 | 21.80 | 29.93 | 6.69 | 256.89 | 57.39 | 63.29 | 14.12 |
| References | China | Beijing | 346.07 | 99.07 | 28.6 | 31.23 | 9.02 | 180.10 | 52.04 | 35.67 | 10.31 |
| | | Shanghai | 342 | 27.70 | 8.1 | 21.55 | 6.3 | 186.39 | 54.5 | 106.36 | 31.1 |
| | | Chongqing | 540 | 51 | 9.4 | 16 | 3.0 | 348 | 64.4 | 126 | 23.3 |
| | Iran | | 515 | 125 | 24.27 | 40 | 7.77 | 250 | 48.54 | 100 | 19.42 |
| | Denmark | | 400 | 96 | 24 | 32 | 8 | 196 | 49 | 76 | 19 |
| | Switzerland | | 639 | 23 | 3.6 | 25.56 | 4.0 | 316.31 | 49.5 | 274.13 | 42.9 |
| | Turkey | | 183 | 18.3 | 10 | 27.45 | 15 | 106.14 | 58 | 31.11 | 17 |

Determination of $S_S$ and $X_S$ Fractions

Using basic respirometry, based on the OUR profile obtained at two F/M ratios of 0.18 and 0.15 g COD·g$^{-1}$ VSS in 20 °C and 10 °C, respectively, for the composite sample, the important COD fraction in the wastewater was evaluated experimentally. As shown in Figure S3, the first section was assigned to the degradation of $X_S$ and $S_S$, both of which were biodegradable. Since $S_S$ degrades much faster than $X_S$, only $X_S$ is present in Section 2 and was further degraded by microorganisms. In the third section, since only endogenous respiration occurred, the values of $S_S$ and $X_S$ could be calculated and regressed from the OUR local curves in the first two parts, respectively.

Using the values of $Y_H$ calculated from Section 3.2.1, the calculated $S_S$ and $X_S$ values of the influent to the CASS process in the triplicate experiments were listed in Table 2. The results showed that the concentration of $X_S$ was higher than $S_S$ both in the summer and winter seasons.

Determination of $X_I$ Fraction

According to Equation (4) and the obtained values of $S_S$, $X_S$, $S_I$ and TCOD from the previous tests, the computed concentration of $X_I$ were 31.33 and 63.29 mg COD·L$^{-1}$ in summer and winter, respectively.

Based on the calculation of the above determined values, Table 2 lists the four COD fractions in different seasons of sewage. These analyzed results demonstrated the following points: (1) Obviously, due to the existence of many particulate fractions, like fine fibers in the CASS influent, the soluble organic substances ($S_S$ and $S_I$) accounted for a little proportion of TCOD; (2) There were some differences in COD components from summer and winter seasons which led to different model parameters. Thus, it was quite necessary to analyze the sensitivity of model parameters in different seasons to get accurate simulation results.

Table 2 also shows the comparison of the proportion of COD components between this WWTP and some other cities and countries. This comparison shows that the wastewater components are greatly affected by the structure and quality of the pipe network and living habits, which proves the necessity of dividing COD fractions. As shown in Table 2, the ratio of $S_S$ and $S_I$ fraction in the studied plant is similar to that in Beijing [69], north of China, while a little higher than that in Shanghai [70] and Chongqing [71], south of China. Compared with other countries, the ratio of $S_S$ fraction in the winter influent of this study is similar to that measured by the Iran sewage plant [19], while the proportion in the summer

influent is slightly higher than that of the Iran sewage plant. The percentage of $X_I$ in the influent of the two seasons is significantly lower than the reported values of Denmark [72], Switzerland [73] and Turkey [74]. In addition, the proportion of $S_I$ and $X_S$ in the plants during the two seasons are also in the scope of literature reports.

### 3.3. Sensitivity Analysis, Calibration and Validation of ASM1 Model

As shown in Figure S4, in summer season, the first three sequences of the most corresponding significant parameters (absolute value) influencing the effluent COD, $NH_4^+$-N and TN were $k_h > \mu_{H,max} > K_S$, $K_{O,H} > \mu_{A,max} > k_h$, and $k_h > \mu_{H,max} > K_S$, while $k_h > K_X > K_{NH}$; $k_h > K_S > b_A$; $K_S > K_X > \mu_{H,max}$ in winter. As a result, it demonstrated that summer parameters $k_h$, $\mu_{H,max}$, $K_S$, $K_{O,H}$, and $\mu_{A,max}$ have the greatest influence on the established ASM1 model for the CASS WWTP, while parameters $k_h$, $K_X$, $K_{NH}$, $K_S$, $b_A$, $\mu_{H,max}$ for winter. As the parameter $f_P$ had little influence on the both the outputs of the summer and winter seasons, the default value was used for the model calibration. After that, the above 16 parameters were selected to further calibrate and verify the ASM1 model applied in the CASS system.

After the sensitivity analysis, the model calibration process was a process involving adjusting the model coefficient values.

Therefore, the results produced by the model using these coefficients are in good agreement with a set of measurements. The calibration process requires preliminary guesses and the logical domain of each coefficient. Such initial values are obtained from the literature [28,30]. In this study, the model has been calibrated for COD, $NH_4^+$-N and TN removals. For two different seasons, nine highly relevant parameters were varied on basis of the causality of the parameters on COD, $NH_4^+$-N and TN, respectively. The selection of calibration parameters was based primarily on the results of the sensitivity analysis.

Table 3 shows the kinetic and stoichiometric parameters that are most suitable for the simulated calibration cycle. The simulated and measured values of the basic output variables characterizing the quality of wastewater are shown in Figure 3. Subsequently, validation of the ASM1 model was conducted during a continue month (July in summer 2018 and February in winter 2019, respectively) of the CASS operation period.

**Table 3.** Kinetic and stoichiometric parameters for ASM1 modelling at 10 °C and 20 °C in this study.

| Parameters | Default Value * | | Calibration Value | | Units |
|---|---|---|---|---|---|
| | 20 °C | 10 °C | 20 °C | 10 °C | |
| **Stoichiometric parameters** | | | | | |
| $Y_A$ | 0.24 | 0.24 | 0.24 | 0.24 | g cell COD formed (g N oxidized)$^{-1}$ |
| $f_P$ | 0.08 | 0.08 | 0.08 | 0.08 | dimensionless |
| $i_{XB}$ | 0.086 | 0.086 | 0.086 | 0.086 | g N (g COD)$^{-1}$ in biomass |
| $i_{XP}$ | 0.06 | 0.06 | 0.06 | 0.06 | g N (g COD)$^{-1}$ in endogenous mass |
| **Kinetic parameters** | | | | | |
| $\mu_{H,max}$ | 6.0 | 3.0 | 9.86 | 3.6 | d$^{-1}$ |
| $K_S$ | 20.0 | 20.0 | 23.72 | 14.86 | g COD m$^{-3}$ |
| $K_{O,H}$ | 0.20 | 0.20 | 0.19 | 0.20 | g O$_2$ m$^{-3}$ |
| $K_{NO}$ | 0.50 | 0.50 | 0.5 | 0.5 | g NO$_3$-N m$^{-3}$ |
| $\eta_g$ | 0.8 | 0.8 | 0.8 | 0.8 | dimensionless |
| $\eta_h$ | 0.4 | 0.4 | 0.4 | 0.4 | dimensionless |
| $k_h$ | 3.0 | 1.0 | 4.1 | 1.3 | g slowly biodegradable COD (g cell COD·d)$^{-1}$ |
| $K_X$ | 0.03 | 0.01 | 0.02 | 0.02 | g slowly biodegradable COD (g cell COD)$^{-1}$ |
| $\mu_{A,max}$ | 0.80 | 0.30 | 0.65 | 0.30 | d$^{-1}$ |
| $K_{NH}$ | 1.0 | 1.0 | 0.95 | 1.0 | g NH$_3$-N m$^{-3}$ |
| $k_a$ | 0.08 | 0.04 | 0.08 | 0.04 | m$^3$ COD (g·d)$^{-1}$ |
| $b_A$ | 0.15 | 0.05 | 0.15 | 0.04 | d$^{-1}$ |
| $K_{O,A}$ | 0.4 | 0.4 | 0.4 | 0.4 | g O$_2$ m$^{-3}$ |

| Parameters | Default Value * | | Calibration Value | | Units |
|---|---|---|---|---|---|
| | **20 °C** | **10 °C** | **20 °C** | **10 °C** | |
| Measured parameters ** | | | | | |
| $Y_H$ | 0.67 | 0.67 | 0.66 | 0.51 | g cell COD formed (g COD oxidized)$^{-1}$ |
| $b_H$ | 0.62 | 0.20 | 0.51 | 0.28 | d$^{-1}$ |

* Default ASM1 value. ** The measured values were not changed during the model calibration.

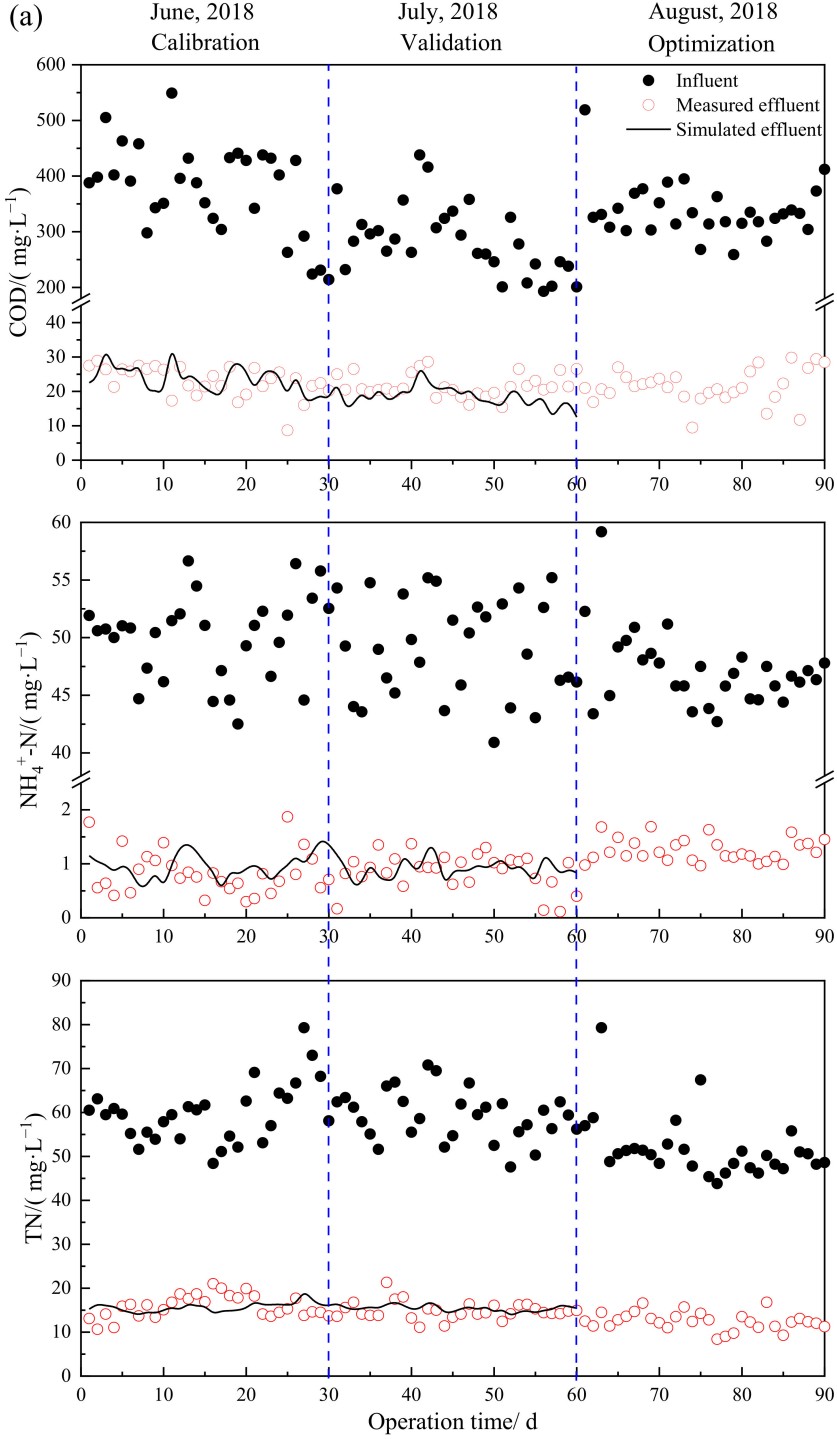

**Figure 3.** *Cont.*

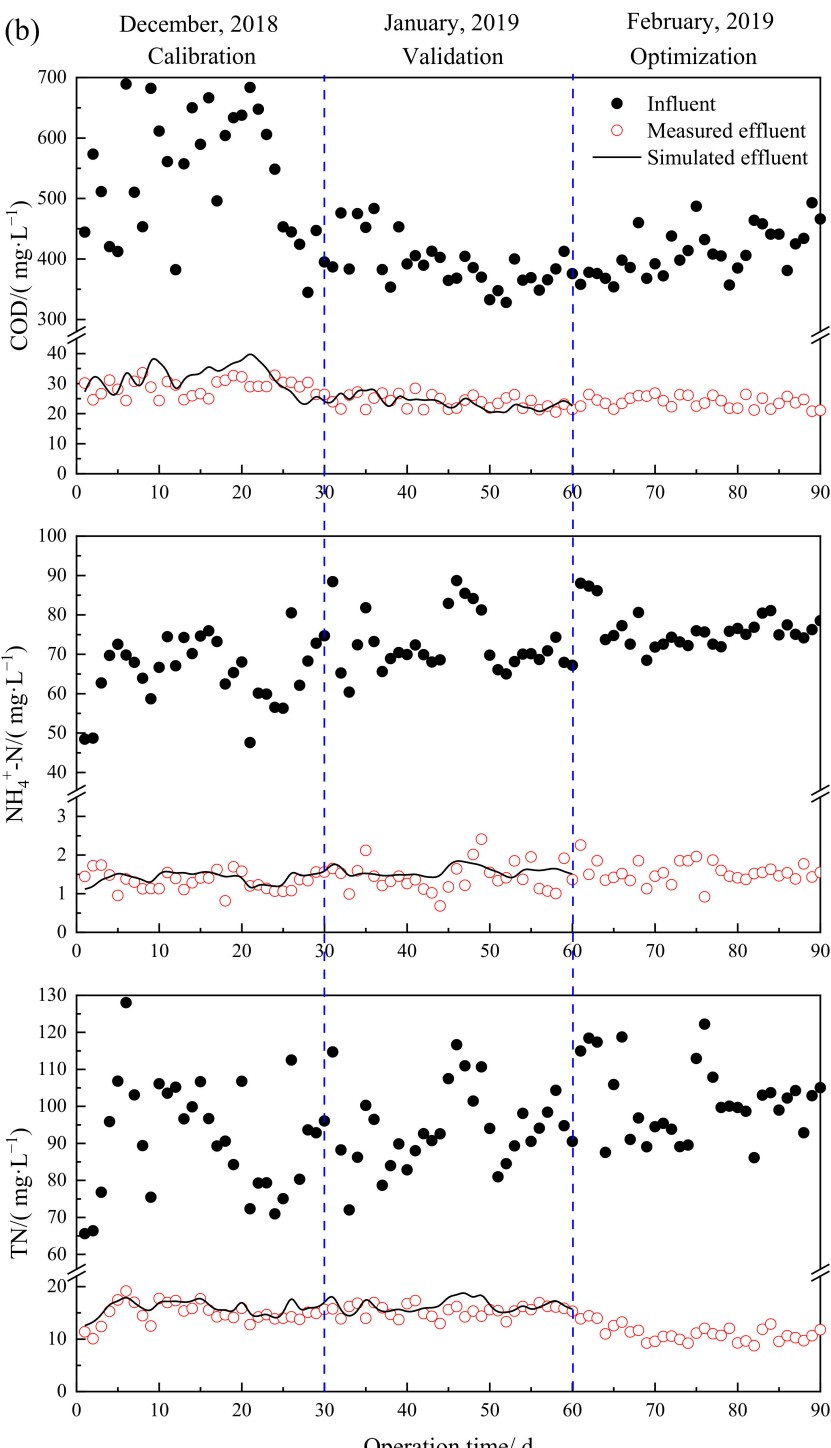

**Figure 3.** ASM1 simulations, validations and optimizations of the CASS WWTP effluent concentrations in COD, NH$_4^+$-N and TN in (**a**) summer and (**b**) winter season.

Regarding the effluent COD, NH$_4^+$-N and TN concentrations as output, the ASM1 model has been running according to the real-time characteristics of daily influent wastewater. As shown in Figure 3, the values obtained in the model predictions and actual measurements are in good agreement. In the summer simulation, the average absolute relative error between the measured and simulated values was 18.51%, 14.72 and 13.73%, with respect to the effluent COD, NH$_4^+$-N and TN, respectively. Similarly, in winter simulation, the average absolute relative error was 15.73%, 18.68% and 9.92%, respectively. Compared with the ideal conditions in the model, this difference may be due to the complex conditions

of the CASS wastewater treatment plant. However, the predicted COD, $NH_4^+$-N and TN concentrations are closely related to the measured concentrations. This shows that the mechanism model can be used to effectively simulate the operation of a full-scale sewage treatment plant. Therefore, the mechanism model is used to generate wastewater quality data under different operating conditions for subsequent optimization.

### 3.4. Optimization of Nitrification and Denitrification Process

As the residence time of the pre-denitrification anoxic zone is too short to meet the discharge standard of nitrogen, aerobic tank was used as a large anoxic zone to extend denitrification time to make full use of influent carbon sources. The first plan of the upgrade solution for this CASS WWTP was to add four submersible agitators in the aerobic pool based on the CFD simulation. The operation strategies were optimized based on ASM1 model.

### 3.4.1. Optimization of the Flow Field

The ASM models assume that the bioreactor of the CASS system is a CSTR unit. However, the actual wastewater plant has uneven mixing due to the improper installation of some devices. So before simulating the nutrient removal process, it is necessary to get a uniform flow field with less short flow or dead zone. In this section, two kind of installation locations were simulated in order to improve hydrodynamics (Figure 1). One way is the traditional installation mode, in which four agitators were installed at four corners of the tank with the installation angle of 45°, respectively. The other way is that four agitators were installed at the same side of the tank with different angles (60°, 45°, 30° and 90°, respectively). The parameters of the agitators using in the CFD simulation was shown in Table S1.

In order to describe the fluid mechanics in the aeration tank after installing the agitator in more detail, the fluid velocity fields of different configurations are shown in Figure 4.

In Figure 4a, low liquid speed in the tank without agitators installed, and uneven mixing in the flow field is not conducive to mixing contact between pollutants and microorganisms. Figure 4b,c show the fluid velocity field of the traditional and novel installation modes of the agitators in the bioreactor of the CASS system. Both modes can realize a good mix of pollutants with microorganisms. However, the flow velocity of the traditional mode is much larger than that of the novel mode. In addition, the larger agitation speed will cause oxygen to enter the wastewater body, which is not conducive to denitrification process. From this perspective, the novel mode is recommended in the upgrading of this wastewater treatment.

### 3.4.2. Optimization of Operation Strategies

The obtained ASM1 was used to optimize operation conditions of the CASS process in the WWTP. In order to accurately control the effluent quality of pollutants and save energy consumption, six cycle operation scenarios (Figure 5) were evaluated to investigate the performance of CASS among various periods of filling, reaction (including mixing and aeration), and settling phases in the batch cycle time with respect to summer and winter, respectively. The influent characteristics were kept the same as Table 1.

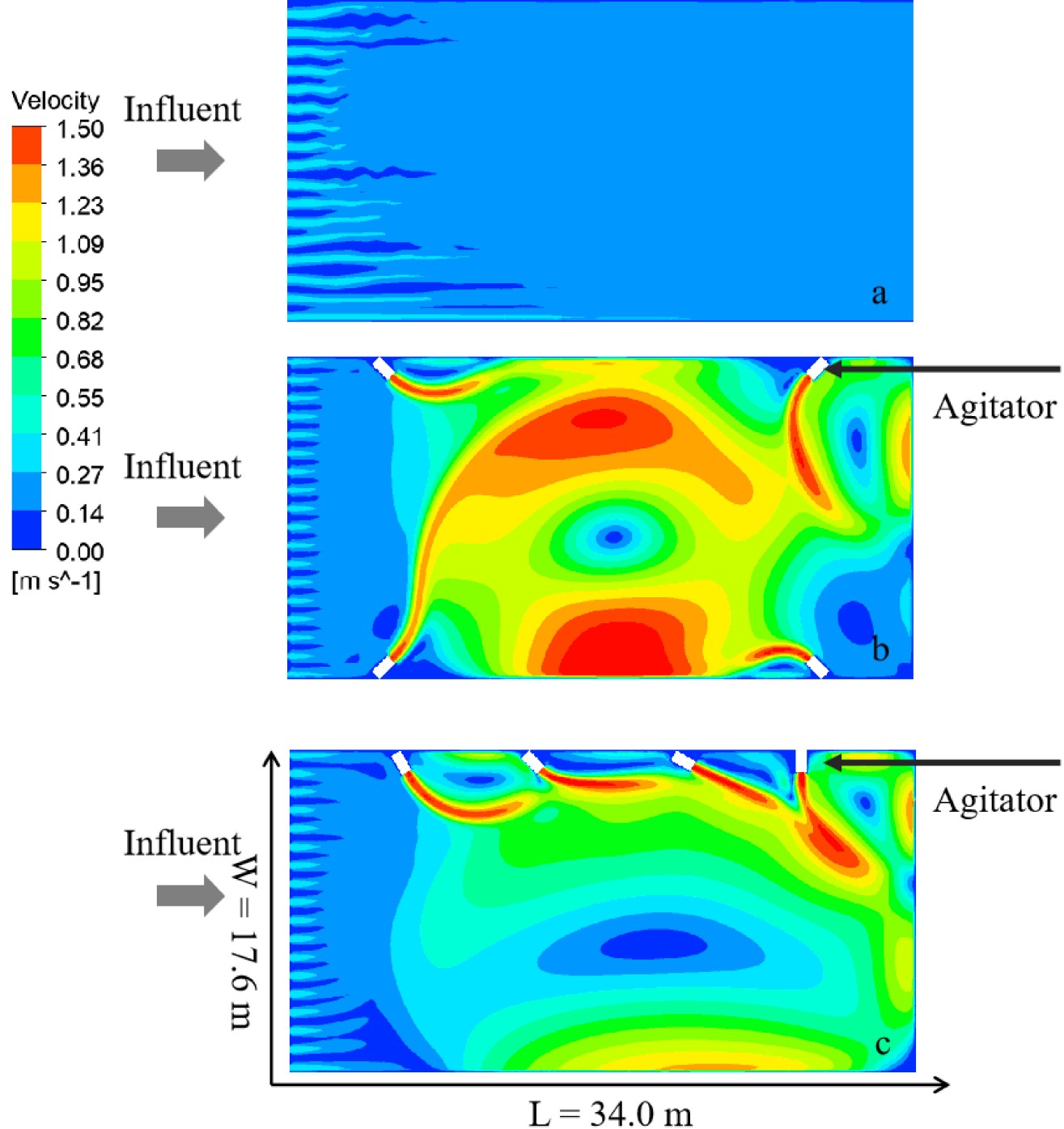

**Figure 4.** Fluid velocity field in aerobic tank of the CASS system: (**a**) No agitators installed; (**b**) Traditional installation mode; (**c**) Novel installation mode.

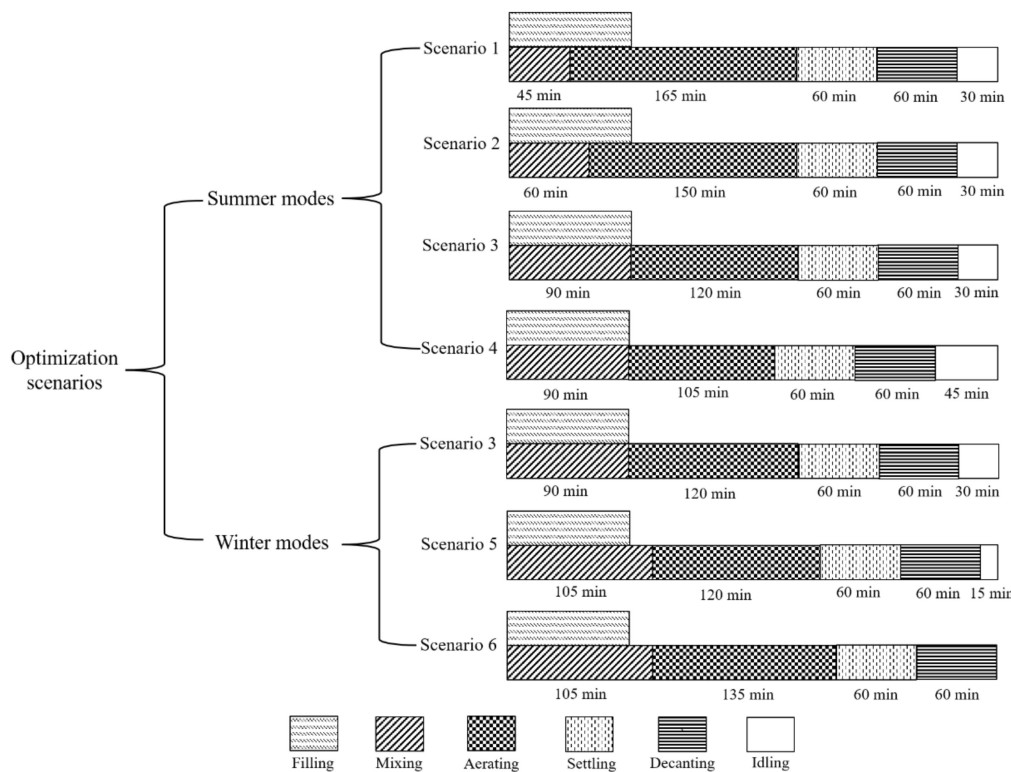

**Figure 5.** The schematic of season cycle operation strategies to be optimized.

During filling, the influent wastewater is mixed with the biomass left in the tank in the previous cycle. During the mixing and filling process, bacteria will biodegrade organic matter and use residual oxygen or other electron acceptors, such as $NO_3^--N$. The purpose of the optimization is to reduce the nitrate in the wastewater through the biological denitrification process to convert the nitrate in the wastewater into nitrogen. Table 4 summarizes the results of the summer and winter operation schemes 1 to 6. In summer scenarios, the TN concentration in effluent decreased with the increase of the mixing time. Only Scenario 3 was suggested, which can meet with the Grade I-A requirements of the discharge standard GB18918–2002. As for Scenario 4, the effluent $NH_4^+-N$ (2.56 mg·L$^{-1}$) increased due to the shorter of aeration time, and the TN in the effluent (15.69 mg·L$^{-1}$) did not meet with Grade I-A standard of GB18918–2002. In winter mode, because of the difference of influent characteristics and water temperature, the COD and TN concentrations in scenario 3 can no longer meet the discharge standard. As the stirring and aeration time was prolonged, the concentration in effluent reduced. Scenario 6 was recommended for the winter operation.

**Table 4.** Simulation results of multiple cycle operational scenarios in this study.

| Scenarios | | Effluent Quality/(mg·L$^{-1}$) | | | Removal/% | | CASS Tank | |
|---|---|---|---|---|---|---|---|---|
| | | COD | NH$_4^+$-N | TN | Nitrification | Denitrification | MLSS mg·L$^{-1}$ | MLVSS mg·L$^{-1}$ |
| Summer modes | 1 | 34.29 | 1.47 | 21.68 | 96.96 | 61.94 | 3250 | 2215 |
| | 2 | 31.76 | 2.17 | 19.76 | 95.52 | 65.32 | 3560 | 2330 |
| | 3 | 24.07 | 1.23 | 14.56 | 97.46 | 74.44 | 3910 | 2850 |
| | 4 | 26.84 | 2.56 | 15.69 | 94.71 | 72.46 | 3780 | 2470 |
| Winter modes | 3 | 66.89 | 4.78 | 16.74 | 91.90 | 81.54 | 4200 | 2660 |
| | 5 | 50.08 | 5.24 | 14.84 | 92.72 | 83.64 | 4520 | 3130 |
| | 6 | 38.54 | 3.27 | 14.17 | 95.52 | 84.38 | 4680 | 3090 |
| Grade I-A of GB18918–2002 | | 50 | 5 (8)* | 15 | | | | |

* Limits of NH$_4^+$-N concentration in effluent are 5 mg·L$^{-1}$ at water temperature >12 °C and 8 mg·L$^{-1}$ at water temperature <12 °C, respectively.

### 3.5. Treatment Performance and Energy Consumption after Upgrading WWTP

3.5.1. Treatment Performance

According to the optimized scenarios for summer and winter, the WWTP in August 2018 and February 2019, respectively, was operated in accordance with the strategies, and the actual treatment performance was shown in Figure 3. The effluent quality of COD, $NH_4^+$-N and TN all met the discharge standard of Grade I-A. After optimization of the cycle operation time, it could be concluded that the ASM1 model can provide improvement measures for wastewater treatment plant accurately and efficiently.

3.5.2. Energy Consumption

The total energy consumption for wastewater treatment before and after upgrade of the WWTP is shown in Figure S5. In summer, after applying the scenario 3, the energy consumption decreased from 0.40 to 0.30 kW·h·m$^{-3}$, which reduced energy consumption by 25%. In winter, the energy consumption was reduced by 16.67% from 0.42 to 0.35 kW·h·m$^{-3}$ after applying the Scenario 6. The reduction of energy consumption in both summer and winter contributed to the reduction of the time of aeration. In a word, both the optimized scenarios not only improved the nitrogen removal efficiency, but also decreased the energy consumption on the premise of ensuring the effluent quality reaches the discharge standard.

### 3.6. Evolution of Microbial Community in CASS

The 16S rRNA gene pyrosequencing was performed to characterize the microbial communities' evolution before and after WWTP modification in summer and winter, with four samples being obtained from the treatment plant. Table 5 shows the alpha index of microbial diversity.

**Table 5.** Alpha index of microbial diversity before and after WWTP modification in summer and winter.

| Scheme | Shannon Index | ACE Index | Chao Index | Coverage Index | Simpson Index |
|--------|---------------|-----------|------------|----------------|---------------|
| Summer original | 5.0814 | 2504.50 | 2522.75 | 0.9913 | 0.02518 |
| Summer optimized | 5.4416 | 2771.51 | 2746.00 | 0.9910 | 0.02301 |
| Winter original | 5.1272 | 2669.78 | 2835.68 | 0.9606 | 0.02297 |
| Winter optimized | 5.5453 | 3643.59 | 3126.00 | 0.9873 | 0.01541 |

The results of the bacteria at the phylum level for each sample are shown in Figure 6a. According to Figure 6a, the predominant phyla in four samples were *Proteobacteria*, *Bacteroidetes* and *Firmicutes*. In summer, before the upgrading, the relative abundances of *Proteobacteria* and *Bacteroidetes* in the microbial community accounted for 44.82% and 24.08%, respectively, similar to the results of bacterial communities in full-scale WWTPs [51,75], and the relative abundances of *Firmicutes* contained a small amount (5.61%). After upgrading, though some increase in *Proteobacteria* (53.85%) and decrease in *Bacteroidetes* (21.44%) occurred, they still dominated in bacterial community. A certain amount of *Verrucomicrobia* (5.79%) appeared in summer. After the upgrading of the plant, the microbial biodiversity of the activated sludge further increased, the dominant population also increased, the overall structure became more balanced. There was a certain similarity in winter sludge community structure change. The relative abundance of *Proteobacteria*, which have both denitrification and phosphorus removal capabilities, increased 5.66% (from 53.10% to 56.02%) after winter upgrading. The relative abundance of *Acidobacteria*, *Firmicutes* and *Acinobacteria* increased 2.24%, 2.92% and 2.15%, respectively. The nitrification capacity of the system has been improved, which is reflected in the decrease in $NH_4^+$-N concentration in the effluent.

(a)

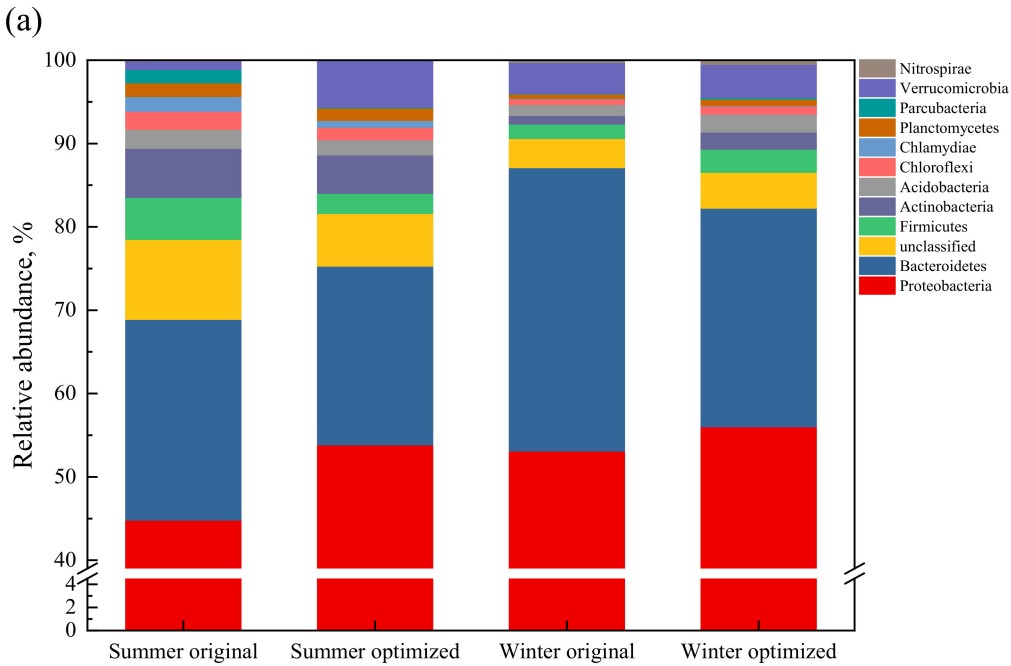

(b)

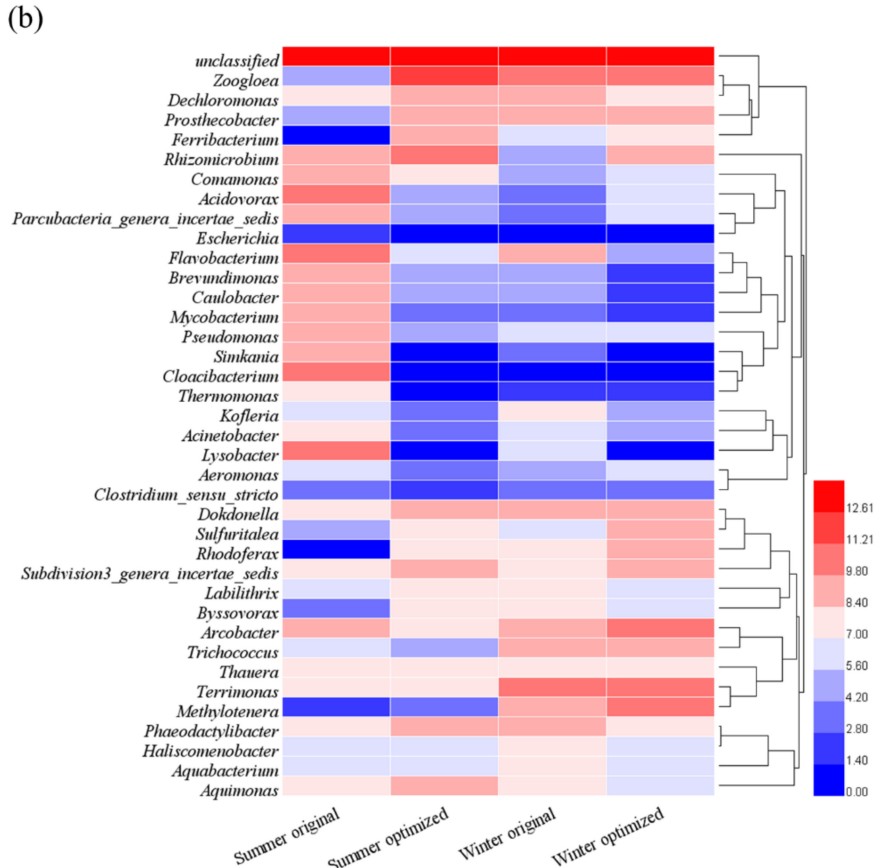

**Figure 6.** The relative abundance of bacterial communities at the (**a**) phylum level and (**b**) genus level (based on the log2 transformed) before and after the upgrading CASS process in different seasons.

Comparing summer with winter, after upgrading, the relative abundance of *Proteobacteria* increased from 53.85% to 56.02%, which shows that the decrease in water temperature has caused the emergence of more *Proteobacteria* in the system that can maintain stable operation of the activated sludge system and release phosphorus and denitrification in the anaerobic and anoxic phases, respectively..

At the genus level (Figure 6b), the genera related to denitrification mainly include *Methylotenera*, *Zoogloea*, *Dechloromonas*, *Arcobacter*, *Thauera*, and *Pseudomonas*. In winter, *Methylotenera* (8.96% to 12.94%), *Zoogloea* (8.66% to 8.74%), *Dechloromonas* (8.15% to 7.63%), *Arcobacter* (8.31% to 9.43%), *Thauera* (7.11% to 7.66%), and *Pseudomonas* (5.43% to 5.67%) in the CASS system accounted for 52.06% of the known microbial genus. These results clearly showed that the denitrifying bacteria in the system increased significantly resulting in changes of $\mu_{H,max}$ and $K_S$ listed in Table 3, after the sewage plant was upgraded and stable operation in low temperature.

## 4. Conclusions

In this study, successfully modelling and its application was performed for the full-scale CASS WWTP in summer and winter season of North China. The main conclusions are made as following:

(1) The solution based on modelling of ASM1 and CFD was successfully applied for upgrading the full-scale CASS process WWTP, which resulted in not only the effluent COD, $NH_4^+$-N and TN concentrations meeting with the discharge standard of Grade I-A, but also reducing energy consumption of the WWTP from 16.67% to 25%.

(2) Influent COD fractions and their difference in summer and winter were determined, and the key characteristic parameters ($Y_H$ and $b_H$) of the activated sludge were determined through the respirometry at temperatures of 10 °C and 20 °C, respectively. The agitators' installation layout in the bioreactor of the CASS process was optimized through CFD simulation. These results guarantee the long-term continuation of the ASM1 modelling simulation validity for optimizing operational scenarios of the CASS process in summer and winter.

(3) The microbial biodiversity of the activated sludge is increased after upgrading, and the relative abundance of the denitrifying bacteria increased significantly, revealing the microbiological significance of the biological reaction kinetic parameters, like $\mu_{H,\,max}$, $K_S$, of the ASM1 model.

**Supplementary Materials:** The following are available online at https://www.mdpi.com/2227-9717/9/3/527/s1, Figure S1: Historical operation data of sewage treatment plants in the past year (2017-03-01 to 2018-03-01), Figure S2: Typical profiles of $NH_4^+$-N, $NO_3^-$-N, TN and COD in the main reaction zone during one cycle in continuous feeding CASS. ((a) Summer original, MLSS = 5745 mg/L, T = 21 °C, 03/06/2018; (b) Winter original, MLSS = 6589 mg/L, T = 10 °C, 5/12/2018; (c) Summer optimized, MLSS = 3261 mg/L, T = 23 °C, 19/07/2018; (d) Winter optimized, MLSS = 4719 mg/L, T = 9 °C, 15/02/2019). Symbols: $NH_4^+$-N (★), $NO_3^-$-N (◖), TN (△) and COD (◆), Figure S3: OUR curves of CASS influent of three repeated experiments in 20 °C and 10 °C, Figure S4: Sensitivity analysis of the ASM1 model parameters on the model outputs, Figure S5: Energy consumption comparison of per volume of treated wastewater ($E_V$) (a) and per unit mass of TN removal ($E_N$) (b) between the original condition and ASM1 optimized condition in summer and winter season. Table S1: The parameters and installation modes of the agigators used in the CFD simulation.

**Author Contributions:** Conceptualization, M.L.; methodology, M.C.; software, M.L.; validation, D.Y., and J.Z.; formal analysis, M.C.; investigation, M.L., R.Q., D.Y., M.Y., and J.Z., H.D.; resources, M.L.; data curation, M.L., H.D.; writing—original draft preparation, M.L.; writing—review and editing, Y.W.; visualization, M.L.; supervision, Y.W.; project administration, Y.W.; funding acquisition, Y.W. All authors have read and agreed to the published version of the manuscript.

**Funding:** This work was supported by the Major Science and Technology Program for Water Pollution Control and Treatment (2017ZX07102; 2015ZX07203–005), the Joint Research Program of National

Natural Science Foundation of China and National Science Foundation of Sri Lanka (NSFC-NSF SL) (21861142020), Program of China-Sri Lanka Joint Center for Water Technology Research and Demonstration by the CAS.

**Institutional Review Board Statement:** Not applicable.

**Informed Consent Statement:** Not applicable.

**Data Availability Statement:** Not applicable.

**Conflicts of Interest:** The authors declare no conflict of interest.

## Abbreviations

| | |
|---|---|
| $A^2/O$ | Anaerobic, Anoxic and Aerobic (WWTP configuration) |
| ASMs | Activated Sludge Models |
| ATU | Allylthiourea |
| $b_A$ | Autotrophic decay rate |
| $b_H$ | Decay coefficient for heterotrophic biomass |
| $BOD_5$ | Biological Oxygen Demand (5 days) |
| CASS | Cyclic Activated Sludge System |
| CFD | Computational Fluid Dynamics |
| COD | Chemical Oxygen Demand |
| CSTR | Continuous Stirred Tank Reactor |
| DO | Dissolved Oxygen |
| EFF | Effluent of WWTP |
| $E_N$ | Energy Consumption per unit mass of TN removal |
| $E_V$ | Energy Consumption per Volume of treated wastewater |
| F/M | The ratio between the SCOD of influent value and the MLVSS, mg COD/mg VSS |
| $f_P$ | Fraction of biomass to particulate products |
| INF | Influent of WWTP |
| $i_{XB}$ | Mass of nitrogen per mass of COD in biomass |
| $i_{XP}$ | Mass of nitrogen per mass of COD in products biomass |
| $k_a$ | Ammonification rate |
| $K_d$ | Global attenuation coefficient |
| $k_h$ | Maximum specific hydrolysis rate |
| $K_{NH}$ | Ammonia half-saturation coefficient for autotrophic biomass |
| $K_{NO}$ | Nitrate half-saturation coefficient for denitrifying heterotrophic biomass |
| $K_{O,A}$ | Oxygen half-saturation coefficient for autotrophic biomass |
| $K_{O,H}$ | Oxygen half-saturation coefficient for heterotrophic biomass |
| $K_S$ | Half-saturation coefficient for heterotrophic biomass |
| $K_X$ | Half-saturation coefficient for hydrolysis of slowly biodegradable substrate |
| MLSS | Mixed Liquor Suspended Solids |
| MLVSS | Mixed Liquor Volatile Suspended Solids |
| OC | Oxygen Consumption |
| OD | Oxidation Ditch |
| OUR | Oxygen Uptake Rate |
| $OUR_{ER}$ | Oxygen Uptake Rate of Endogenous Respiration |
| OURtot | Total Oxygen Uptake Rate |
| $OUR_{Xs}$ | Oxygen Uptake Rate of $X_S$ consumption and endogenous respiration |
| SRT | Sequencing Batch Reactor |
| SS | Suspended Solid in influent |
| T | Time (hour) |
| $V_S$ | Volume of Sludge |
| $V_W$ | Volume of Wastewater |
| $X_{0B,H}$ | Initial concentration of active heterotrophic biomass |
| $Y_A$ | Yield for autotrophic biomass |

| | |
|---|---|
| $Y_H$ | Yield for heterotrophic biomass |
| $\eta_g$ | Correction factor for $\mu_H$ under anoxic conditions |
| $\eta_h$ | Correction factor for hydrolysis under anoxic conditions |
| $\mu_{A,max}$ | Maximum specific growth rate for autotrophic biomass |
| $\mu_{H,max}$ | Maximum specific growth rate for heterotrophic biomass |

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
