# Peer review of "Model-Based Solution for Upgrading Nitrogen Removal for a Full-Scale Municipal Wastewater Treatment Plant with CASS Process"

_processes, doi:10.3390/pr9030527_

Round 1

Reviewer 1 Report

In the reviewed paper an approach for upgrading the biological stage of the existing full scale municipal WWTP with CASS SBR process is presented. The main objective of the upgrade was to improve effluent quality to meet current discharge limits, especially in terms of nitrogen. Optimization was carried out with the use of computer modelling. For this purpose GPS-X programme with ASM1 model was employed. Additionally, CFD modelling was conducted which resulted in changes to the agitator installation. The results presented relate not only to treatment performance, but include microbiological analysis using high-throughput 16S rRNA gene sequencing.

The paper is an interesting and valuable piece of work, also in the aspect of practical application. In my opinion it is suitable for publication in Processes. However, I have a few comments.

The English language is generally correct, but some sentences are very long and therefore less understandable. Moreover, some phrases need to be revised. Below are some examples of language or other small deficiencies noticed.

Line 101: no space in „0.5mg/L”

Line 106: an abbreviation for WWTPs is explained, but it is not the first appearance as “WWTPs” is mentioned earlier in the text (line 94)

Line 107: no space in “50,000m3/d”

Lines 109-112: the first part of the sentence concerning two kinds of WWTPs could be improved in terms of style

Line 133: “these models” appear, but no models were mentioned earlier

Line 135: please check the current owner of WEST

Lines 150-151: the part “one of the major limitations for a more widespread application of ASM to choose a set of related parameters” could be improved

Line 152: wrong reference (first names used)

Line 158: the sentence ends with “which either promote or inhibit”, could be supplemented by specifying what is promoted or inhibited

Line 179: extra brackets inside references

Line 216: the sentence starting with “And…” could be improved in terms of style

In table 1 flow rate values have oddly placed comma (four digits from the right?)

Lines 262-264: the style of the sentence could be improved (e.g. “the reflux ratio for the influent with returned sludge”)

Line 290: “samples taken (…) of each summer and winter in 2018“ – oddly worded

Line 295: F/M is mentioned, maybe it could be added to the abbreviations list at the beginning

Line 304: the number is missing in “After increasing above mg·L-1”

Lines 304-306: these two sentences would sound better in the passive voice

Line 324: unnecessary capital letter in the word  “where”

Line 330: “In the OUR test of bH,”?

Line 342: “and it was further measured OUR” – oddly worded

Line 343: “pH and temperature were controlled constant corresponding values for summer and winter” – the style could be improved

Line 345: subsection 2.2.2?

Line 358: “true solution COD” ?

Line 368: „the OUR value was in sharply decrease as a results of the substrate was easily biodegradable” – grammar could be improved

Line 416: in the reference Petersen is not the only author

Lines 426-429: style of the sentence could be improved, maybe “(…) which was used as the CFD modelling”

Lines 431-433: the style could be improved in the sentences about discrete spatial derivatives and discrete pressure as something is missing now

Line 441: “in the order of DNA extraction” or maybe “in order to perform DNA extraction”?

Line 442: something is missing in the sentence starting from “Amplification…”

Line 443: would sound better in the passive voice

Line 470: “consumed a large amount of aeration energy consumption” – oddly worded

Line 472: “nitration”?

Line 485: unnecessary “which” word

Line 504: style could be improved

Lines 513 and 519: please check the section numbers

Lines 517-523: style could be improved by splitting the sentence into parts

Line 557: “that” is unnecessary

Line 595: a typo in “fig. 3”

In figure 6 “(b)” is missing

Generally, the paper is well organized. The methods section is described in detail and the consecutive steps taken during the study are clearly explained. The results section contains the comparison with other papers. I only have a few minor comments and questions.

Lines 230 and 233: there is a large difference in the given heights of anoxic and aerobic zones?

Line 251: Could you please explain why there was no idling phase since the aeration phase was shortened to 150 min?

Line 335: in Equation (2) the parameters except for “fvC” are not explained

Line 349: SI fraction is not biodegradable

Lines 383 and 384: in Equations (5) and (6) newly introduced parameters could be explained

Line 386: version of the GPS-X software used could be added

Line 468: it is stated that “the residual water in the watershed diluting the incoming water”, and is it not the case that the reactor content is diluted by the incoming water?

Lines 470-473: the description of the processes taking place is a little unclear, e.g. the incorporation of HRT. Besides, it is stated that “The rapid decline of NH4+-N occurred obviously when the COD concentration was below 40 mg·L-1.” This is true only for summer (figure S2a), but it can also be observed that at the same time the filling process was ended. It is different in winter, however, when the rapid decrease in ammonia concentration started with the aeration turned on, even though the COD concentration was high and decreasing.  In addition, the aeration phase indicated in figure S2b is a little confusing, as if the phase was beginning after the filling process, not mixing.

Lines 549-550: point (1) is unclear, could you please explain it?

In table 2 the values shown for Switzerland (especially Ss) are very usual.

In table 3 please check the calibrated value for heterotrophic yield. And what do the asterisks mean?

In fig. 3 (b) is this the correct year (2018) for validation and optimization data?

Author Response

Sorry for the poor English writing, and thanks for your positive and constructive comments on this work, it means so much to this paper. Typos and awkward expressions in the manuscript are carefully corrected according to your and the other reviewer’s suggestions and comments, and the language grammar and vocabulary are crossed checked with the help of an expert (Professor) and a native English speaker. They are listed below and marked in red in the manuscript. Please see the attachment.

Reviewer 2 Report

This paper modelled the upgrading nitrogen removal for a full-scale municipal wastewater treatment plant with CASS process. The Influent COD fractions and the key characteristic parameters of the activated sludge were determined. This paper can be accepted after the following revisions:

  1. Please try not to use the abbreviations in Abstract. The full term of the abbreviations should be given in Abstract or the main text such as WWTP.
  2. Please unify the font size of the words in the text.
  3. This paper is based on the ASM1 and CFD models. What are your major improvements compared with others using the same model?
  4. In Introduction, the authors mainly introduced the status of wastewater treatment in China. I think that the authors should introduce the global developments in this field before the China case study. Moreover, the different wastewater treatment methods should be discussed not only the biological approach. Some related works should be considered in the text to show the advances of your method (Modelling demand response with process models and energy systems models: Potential applications for wastewater treatment within the energy-water nexus; A review on agro-industrial waste (AIW) derived adsorbents for water and wastewater treatment). How could you evaluate the economics of the various methods for wastewater treatment?
  5. Please reduce the length of the last paragraph in Introduction. This paper is used a Chinese standard of GB 18918-2002. How could you evaluate the results with an International standard?
  6. Since this work is mainly based on a modeling work, please compare the advantages of modeling with the experimental studies.
  7. In Section 2.1, how did you choose this plant as the representative one?
  8. Fig. 1a is not clear to show the schematic of this process.
  9. In the analysis, 10 and 20 °C were considered in Fig. 2. What is the effect of a higher temperature if the planted is located in the south?
  10. In Fig. 3, all the data presented are in 2018. Will you have the predicted results after 2020 since it is 2021 now?
  11. Please add the coordinate axis in Fig. 4. 
  12. The verification of the modeling results should be highlighted in the conclusions. How could you use the current models for other plants?
  13. Please also improve the English writing significantly in the revision.

Author Response

Sorry for the poor English writing, and thanks for your positive and constructive comments on this work, it means so much to this paper. Typos and awkward expressions in the manuscript are carefully corrected according to your and the other reviewer’s suggestions and comments, and the language grammar and vocabulary are crossed checked with the help of an expert (Professor) and a native English speaker. They are listed below and marked in red in the manuscript.

Reviewer 3 Report

The manuscript ID processes-1092057 is interesting and well written. Meets the standards of scientific paper and technical elaboration. In my opinion, it fully corresponds to the Processes journal profile. Authors have described cost-effective approach and application for upgrading the existing municipal WWTPs in cold region of China to meet more stringent discharge standard of nitrogen removal. This is an important issue from a practical point of view and concerns many regions with low temperatures in winter. This is a very interesting and important direction of research with a very high application potential. In my opinion, the manuscript can be published as it is. Great job. Congratulations !

Below my remarks for consideration by the authors:

  1. Graphical abstract is illegible and should be improved. 2.
  2. The source of funding should be stated.
  3. According to instruction for authors references must be numbered in order of appearance in the text (including table captions and figure legends) and listed individually at the end of the manuscript.

Author Response

In my opinion, the manuscript can be published as it is. Great job. Congratulations!

Below my remarks for consideration by the authors:

  1. Graphical abstract is illegible and should be improved. 2.
  2. The source of funding should be stated.
  3. According to instruction for authors references must be numbered in order of appearance in the text (including table captions and figure legends) and listed individually at the end of the manuscript.

Response:Thanks so much for your approval. We have change some formats according to the instructions for authors.

Round 2

Reviewer 2 Report

I do not think that the authors have answered the questions very well. Please address these questions in the revised paper.

  1. In Introduction, the authors mainly introduced the status of wastewater treatment in China. I think that the authors should introduce the global developments in this field before the China case study. Moreover, the different wastewater treatment methods should be discussed not only the biological approach. Some related works should be considered in the text to show the advances of your method (Modelling demand response with process models and energy systems models: Potential applications for wastewater treatment within the energy-water nexus; A review on agro-industrial waste (AIW) derived adsorbents for water and wastewater treatment). How could you evaluate the economics of the various methods for wastewater treatment? -- Please compare these methods in the paper. The global status also needs to be discussed in the paper. 
  2. The English writing did not have any improvements. Please improve the English writing carefully. Please highlight these changes in the text.
  3. In Fig. 3, all the data presented are in 2018. Will you have the predicted results after 2020 since it is 2021 now? -- Please add the latest data in the paper. If not, I do not think that this paper has much importance to this area especially only a China case study.

Round 3

Reviewer 2 Report

Please improve the English writing significantly before the acceptance.